# Calcined Hydroxyapatite with Collagen I Foam Promotes Human MSC Osteogenic Differentiation

**DOI:** 10.3390/ijms23084236

**Published:** 2022-04-11

**Authors:** Veronika Hefka Blahnová, Lucy Vojtová, Veronika Pavliňáková, Johana Muchová, Eva Filová

**Affiliations:** 1Deparment of Tissue Engineering, Institute of Experimental Medicine of the Czech Academy of Sciences, 14220 Prague, Czech Republic; veronika.blahnova@iem.cas.cz (V.H.B.); eva.filova@iem.cas.cz (E.F.); 2Department of Biophysics, Second Faculty of Medicine, Charles University, 15006 Prague, Czech Republic; 3Advanced Biomaterials Group, CEITEC—Central European Institute of Technology, Brno University of Technology, 60177 Brno, Czech Republic; veronika.pavlinakova@ceitec.vutbr.cz (V.P.); johana.muchova@ceitec.vutbr.cz (J.M.)

**Keywords:** collagen, bioceramics, osteogenesis

## Abstract

Collagen I-based foams were modified with calcined or noncalcined hydroxyapatite or calcium phosphates with various particle sizes and pores to monitor their effect on cell interactions. The resulting scaffolds thus differed in grain size, changing from nanoscale to microscopic, and possessed diverse morphological characteristics and resorbability. The materials’ biological action was shown on human bone marrow MSCs. Scaffold morphology was identified by SEM. Using viability test, qPCR, and immunohistochemical staining, we evaluated the biological activity of all of the materials. This study revealed that the most suitable scaffold composition for osteogenesis induction is collagen I foam with calcined hydroxyapatite with a pore size of 360 ± 130 µm and mean particle size of 0.130 µm. The expression of osteogenic markers RunX2 and ColI mRNA was promoted, and a strong synthesis of extracellular protein osteocalcin was observed. ColI/calcined HAP scaffold showed significant osteogenic potential, and can be easily manipulated and tailored to the defect size, which gives it great potential for bone tissue engineering applications.

## 1. Introduction

The incidence of various bone pathologies is rising worldwide, in common with the number of people with fractures caused by trauma. Even though bone has the capacity to regenerate, fracture healing is a very complex process influenced by a variety of factors. Under optimal physiological conditions, factures heal within a few weeks. Complications occur in approximately 10% of cases when tissue restoration is too slow or has no presence [1]. A typical example of a badly healing injury is a critical-sized bone defect, defined by its dimensions. Moreover, this defect will not heal spontaneously within the patient’s lifetime [2]. An autologous bone graft is considered to be the gold standard of bone grafting material. It is highly advantageous, as it is nonimmunogenic and histocompatible. It also contains living bone cells, which can respond to stimuli from the surrounding microenvironment. On the other hand, autograft harvesting is associated with donor site morbidity, blood loss, and possible chronic pain [3]. Furthermore, its resorbability rate can be so high that the graft is degraded before the new tissue has been restored [4]. Allogenic and xenogenic bone substitutions are also used in certain clinical applications [5]. Their limitations include the possibility of viral transmission and immune system reaction induction. Therefore, prolonged healing time after these types of graft implantation has been observed. Thus, there is an effort to develop a synthetic scaffold usable for bone graft substitution. The material used must be biocompatible, not cytotoxic or carcinogenic, and must not influence physiological body processes in a negative way. Beneficial features are biological activity, osteoinductivity, and osteoconductivity, enabling cell growth and spreading and subsequent bone formation [6].

Nowadays, we have a wide selection of materials to prepare artificial scaffolds [7]. Their mechanical and morphological properties and solubility can be modified by choosing certain fabrication processes, chemical optimization, or component ratio. The main objective is simply to imitate the natural microenvironment that surrounds the cells. Therefore, in bone tissue engineering there is great interest in calcium phosphate compounds and collagen type I as components of the bone tissue. The apatite crystals impregnate collagen fibers and ensure resilience and stiffness of the bone tissue, whereas the organic part provides flexibility [8]. The most common ceramics in use are hydroxyapatite (HAP)- and calcium phosphate (CaP)-based [9,10,11,12,13]. In the literature, there are many outstanding studies dealing with collagen and phosphates scaffolds [14,15]. On the other hand, there are no comprehensive studies comparing different types of phosphates in the same scaffold matrix and concentration, as well as using the same MSC lines. The articles deal only with individual calcium phosphates, but it is not clear which of them is suitable, e.g., for in vivo testing and further for clinical practice.

In our study, we combined six different types and forms of hydroxyapatites (calcined and noncalcined hydroxyapatite with different particle sizes) and calcium phosphates (α- and ß-tricalcium phosphates and finally calcium polyphosphate) with collagen type I, as the main organic part of bone [16]. Hydroxyapatite is a synthetic biomaterial with the chemical formula Ca_10_(PO_4_)_6_(OH)_2_. A range of various techniques has been developed to fabricate HAP particles, when one of the parameters represents the calcination temperature. While noncalcined HAP forms a needle-like morphology, the calcination procedure results in a rod-like morphology of the HAP; moreover, the crystallinity and the particle size increase with increasing calcination temperature [17] It has also been observed that the HAP synthesized at 1000 °C has a similar stoichiometry ratio (Ca/P = 1.65) to that of natural bone [18]. Lyu et al. in their work studied the effects of particle size of hydroxyapatite on mechanical property and cytocompatibility of hydroxyapatite/poly(amino acid) composites [19]. Results showed that with increasing HAP particle size, the mechanical properties of the composites are enhanced, and without impacting their cytocompatibility and degradation behavior [19].

α-tricalcium phosphates (α-TCP) have the excellent mechanical strength of hydrated products and a suitable self-hardening capacity, as well as being a biocompatible product of HAP [20]. α-TCP cement paste can be hardened in the presence of body fluid and blood, and effectively adapts to the irregular shape of bone defects [21]. ß-tricalcium phosphates (ß-TCP) are widely used in bone tissue engineering due to their excellent bioresorbability. Therefore, they can be resorbed and replaced by newly formed bone tissue, and there is no need for a second surgical operation to remove the device after healing occurrence [22]. Conversely, β-TCP has been shown to possess weak mechanical properties in load-bearing applications [23]. Polyphosphate (polyP) is a naturally existing polymer composed of orthophosphate units [24] that are present in platelets and in the serum. It has been found that complexed calcium polyphosphate microparticles (CapolyP) improve osteoinductive properties in preclinical studies [25]. PolyP is commonly used in cosmetics and also as a food additive (E 452) [26]. As such, polyP is considered a safe material in human applications. All of the substances differ with respect to their solubility, which allows the desired ratio of biodegradability and bone ingrowth to be set by choosing the chemical composition of the certain scaffold. Despite the fact that bioceramics present significant benefits, such as similarity to the mineral phase of bone, they also incur various disadvantages, e.g., low mechanical toughness or non-homogenous particle size and shape. Furthermore, conventional synthesis leads to grain size at the microscale level, whereas the natural dimensions of the bone extracellular matrix are in nanometers [27]. The production of bioceramics at the nano scale is quite challenging; however, the effort is reasonable. Among other features, they also provide improved biological activity [28] such as cell adhesion, proliferation, and calcium compound deposition.

Therefore, a promising approach is the development of a porous nanostructured bioceramic carrier, combined with the organic phase [29]. It is acknowledged that in natural cortical bone there are pores ranging from 190 to 230 µm, while cancellous bone contains larger pores of approximately 500 to 600 µm [30]. The benefits of larger pores (greater than 300 μm) for cell growth have been experimentally proven in the study of Kuboki [31] and Filová [32]. Meanwhile, the importance of having pores greater than 300 μm for osteogenesis has been shown, too. After all, the topographical features are all very important for cells’ good health. Moreover, the pores create a structural gradient, which is considered to be crucial for osteogenic processes [33]. HAP particles of approximately 80 nm support cell proliferation, while particles of 20 nm are most likely to be captured and gathered by the cultured cells. In this case, the nanoparticles have a cytotoxic effect and decrease the proliferation rate [34]. However, the optimal physicochemical properties are to be determined as the same ideal material for the process of bone regeneration.

In this study, we obtained a porous 3D scaffold (with particle size in the range from 30 nm to 30 μm and pore size in the range from 50 μm to 950 μm) to thoroughly investigate its effect on cell viability, proliferation, osteogenic induction, and osteocalcin synthesis.

## 2. Results

### 2.1. Scaffold Structure and Morphology

The SEM visualizations of each scaffold revealed a ragged surface with many interconnected pores of even distribution (Figure 1). The smallest pore size was measured for the scaffold made of pure collagen (A) and collagen with noncalcined hydroxyapatite (group D) (320 ± 120 µm, 320 ± 110 µm, respectively). In contrast, the biggest pore size was measured for the scaffold made of the combination collagen with hydroxyapatite Reicke (B) (490 ± 170 µm). The pore sizes of the prepared scaffolds are summarized in Table 1. As is apparent, each hydroxyapatite modification had a completely different effect on the scaffold morphology and pore size. Additionally, each of the two tricalcium phosphates led to obvious differences in the pore sizes. In general, the addition of any calcium phosphate caused a morphological change in the collagen scaffold. Pure collagen scaffold is characterized by a honeycomb-like structure, which became more fibrous with the addition of ceramic nanoparticles. All of the samples included pores on a scale of hundreds of nanometers.

### 2.2. A Significantly Higher Level of Metabolic Activity Was Observed in Samples with the Combination of Coli and Hap or Tcp Scaffolds

For the duration of the 21-day experiment, the metabolic activity of the cells grew continuously (Figure 2). Statistically significant differences were observed on days 14 and 21. On the scaffolds made of pure collagen (A) and combined with CapolyP (G), significantly lower levels of metabolic activity were measured in comparison with the remaining experimental groups. The results indicate that mechanical stimulation of the microenvironment was sufficient for the whole period.

### 2.3. A Significantly Higher Amount of dsDNA Was Detected in Samples with the Combination of Coli and Hap or Tcp Scaffolds

The total amount of cellular dsDNA measured by the PicoGreen assay is in strong agreement with the data obtained from the MTS test. Figure 3 shows that cell proliferation was significantly lower on scaffolds made of pure collagen or in combination with CapolyP. In these two samples (A and G), the dsDNA concentration was at approximately the same level on days 1 and 21. The total dsDNA amounts of other samples increased.

### 2.4. RunX2 and ColI mRNA Expression Were Significantly Higher in Pure ColI Scaffold or in Combination with Calcined HAP

As an early marker of osteogenic differentiation, we used the transcription factor RunX2, which regulates all later phases of osteogenesis. The level of mRNA expression was measured on days 1, 14, and 21 (Figure 4). On day 14 significantly higher levels of mRNA expression were reached in all samples than in the negative control group. The statistically highest RunX2 mRNA expression was observed on the scaffold made of Coll and calcined HAP on day 14. In the later phase of our experiment, we observed a decrease in RunX2 mRNA expression in samples A, C, E, F, and G. In contrast, in samples B and D, where the cells were cultured with Coll+HAP Reicke and ColI + noncalcined HAP, respectively, there was an apparent increase in expression, which indicates the process of osteogenesis. This was delayed in comparison with the remaining samples.

As can be seen in Figure 5, higher levels of ColI mRNA expression were reached in all of the experimental samples than in the negative control group. Significantly lower levels were measured on day 21 in the samples cultured on ColI+HAP Reicke, ColI+noncalcined HAP, and, obviously, the negative control group. mRNA expression grew continuously during the whole experiment. Statistically significantly higher levels were detected in samples A (ColI) and C (ColI + HAP calcined). This result is in good agreement with RunX2 mRNA expression, where in the same samples, significantly higher levels of this transcription factor were detected, leading to a subsequent stronger stimulation of ColI mRNA synthesis.

### 2.5. Bone Extracellular Protein Osteocalcin Was Present in High Amounts in the Samples with ColI and HAP Scaffolds

On day 21, initial phases of osteocalcin synthesis were visible in all of the samples except for the cells cultured on pure Coll scaffolds and Coll+CapolyP scaffolds (Figure 6). On day 35, a strong synthesis of osteocalcin was observed in all of the samples containing collagen and all chemical modifications of HAP—Reicke, calcined, and noncalcined. Traces of osteocalcin were also present in samples E and G, where the cells were cultured on ColI and β-TCP and CapolyP, respectively. The density of the nuclei provides good evidence that the cells were confluent by the last experimental day.

### 2.6. The Number of Cells on Scaffolds Increased in All Samples

Photos from confocal microscopy show that on each scaffold, a comparable number of mesenchymal stromal cells were seeded (Figure 7). They were spread evenly. Confocal microscopy on day 14 showed that the number of cells of each scaffold increased. The cells were not aggregated and were still spread evenly.

## 3. Discussion

As cell behavior is directly affected by the scaffold morphology through specific integrin–ligand interactions between cells and their microenvironment, scaffold architecture can influence cell proliferation and control differentiation [35]. The important role of porosity and interconnectivity in also promoting cell migration within the pores and enabling cell proliferation without overgrowing is noteworthy. Pamula et al. [36] synthesized a degradable copolymer of L-lactide and glycolide (PLG) of the same porosity and pore sizes of 600, 200, and 40 µm. The 600 µm pores proved to be the most suitable for MG-63 cell growth. It was also demonstrated that the preferential proliferation for pore sizes is between 250 and 500 μm [37]. The beneficial features of larger pores (greater than 300 μm) for cell growth were also shown in the study of Kuboki [31], Filová [32] and Dušková-Smrčková [38]. It was apparent that the positive effect of small pores on the initial cell attachment was outweighed by the ability of larger pores to allow cell infiltration. Furthermore, this study showed the importance of having pores greater than 300 μm for osteogenesis to occur. The pore sizes of the prepared scaffolds in this study ranged between approximately 100 and 500 μm (Table 1), which is not by any means extreme. As was apparent from our results (Figure 3), the strongest proliferation was present on the carrier with the largest pores (sample F—an average pore size of 460 ± 230 μm). This is in accordance with studies indicating that larger pores are more convenient for cell growth. In experimental samples where the pore size started at 200 μm, the proliferation level was significantly lower. However, these groups did not show a higher cell adhesion on the first experimental day as was reported in the work of Kuboki [31]. The particle size of used calcium phosphates should also be considered when evaluating the cytocompatibility of certain carriers. Studies to date investigating the role of particle size in cell growth have been performed with particles on the nanoscale level [34,39]. The cell growth was shown to have increased on nanoparticles with higher diameter. Particles of approximately 20 nm were most likely captured and gathered by the cultured cells. In this event, these nanoparticles had a cytotoxic effect and decreased the proliferation rate [34]. However, we were working with particles of sizes ranging between 29 nm and 30 μm, which were too big for encapsulation. Therefore, we did not notice a dependence of proliferation on either particle size or on pore size. A significantly decreased level of both cell metabolic activity and proliferation was observed in the pure collagen scaffold as well as the Coll+CapolyP scaffold (Figure 1 and Figure 2). However, in this instance, the significant difference was more likely caused by scaffold material than pore or particle size.

As indicated above, it should be noted that the cell behavior also strongly depends on the fabrication conditions and scaffold material used. Certain materials, typically bioceramics, release calcium and phosphate ions during degradation [40,41,42,43]. This happens both in vitro (water dissolution) and in vivo (by osteoclastic cells). To begin with, calcium ions cause pH changes in their surroundings [44]. It is also known that calcium is an important messenger molecule that triggers various cellular signaling pathways, from which may lead to cell proliferation [45]. Ca^2+^ also participates in the binding of various specific proteins through controlling conformational changes of their partner, predominantly calmodulin [46]. Additionally, phosphorus is a strongly biologically active molecule. It is contained in a wide range of substances such as proteins, nucleic acids, and adenosine triphosphate, and also affects physiological processes [47]. Apart from the ion release, artificial bioceramics are likely to adsorb ions from the culture medium due to their high surface area and lead to significant pH changes [48,49]. This does not cause problems in the body where the fluid circulates and constantly brings fresh ions, but during in vitro cultivation, such a scaffold can be assumed to be cytotoxic. However, any of the problems with pH or dramatic ion adsorption mentioned above were most likely not our issue. Cell viability measured by the MTS test was shown to be growing during our experiment (Figure 2). Significantly lower values on days 14 and 21 were found in the samples cultured on pure collagen scaffolds and Coll+CapolyP foams. These two groups were also found to be less suitable for cell proliferation, as is apparent from the graph (Figure 3) showing the total dsDNA amount. Besides the Coll scaffold, the lowest proliferation rate was reached on the Coll+CapolyP scaffold (Figure 3). This material has the lowest Ca/P ratio and therefore possesses the fastest solubility of all of the materials used [50]. On the other hand, that scaffold was the only group where we noticed an almost two-fold increase in the dsDNA amount from days 1 to 7. This could be caused by an ion burst release during the first days of cultivation and subsequent stimulation of cell growth. While there could be a lack of ions for the cell growth during the last two weeks of cultivation, the Coll scaffold has previously been shown to promote cell growth and metabolic activity [51]. As is apparent from the graphs (Figure 3), the amount of dsDNA remained almost on the same level throughout the whole experiment, and metabolic activity support was only mild.

Although little is known about the ability of collagen/calcium phosphate carriers to induce osteogenesis, various studies have demonstrated that bioceramics can serve as a promising material for grafting applications [10]. It has also been shown that biomaterial porosity and pore size play a very important role in osteogenesis in vitro and in vivo. In vitro, lower porosity suppresses cell proliferation and forces aggregation, resulting in osteogenesis stimulation. According to this work, pores larger than 300 µm are recommended in order to support in vivo capillary formation, which enables oxygen supply and direct bone formation without cartilage pre-stage [52]. However, highly interconnective pores of about 1000 µm in poly(ε-caprolactone) scaffolds strongly enhanced alkaline phosphatase activity, and thus the osteogenesis of human adipose MSCs [53]. On the other hand, a significant osteogenic effect was also induced by using Ti6Al4V foam with an average pore size of 178 µm [54]. All of the results listed above clearly demonstrate that the cell fate is more likely influenced by the material used than by the pore size. We also assume this to be the reason we did not observe any effect of pore size on the level of osteogenesis in our study. However, what we do consider to be crucial for both in vitro and in vivo osteogenic differentiation is the concept of a structural gradient [33]. In accordance with this, various pore sizes should be present in order to provide optimal biomechanical conditions for cell differentiation—the same as in natural bone tissue. Smaller pore sizes provide a large surface for the initial adhesion but, due to extracellular matrix deposition, by quick filling of the pores, the nutrient and oxygen flow can be negatively influenced, whereas larger pores give cells more space for confluent layer formation, aggregation, and matrix deposition, thereby enabling sufficient bioactive molecules and oxygen flow. Larger pores are also advantageous because they allow capillary ingrowth. As is apparent from Figure 1, all our scaffold structures fully met this requirement.

The release of calcium and phosphate ions during scaffold degradation is probably the main reason for CaP biomaterials biological potential [55]. Experimental data clearly demonstrate the crucial role of calcium ions in osteoblast proliferation, differentiation, and activity throughout the natural bone remodeling cycle [56]. It was also shown that the process of human bone marrow MSC osteogenic differentiation is connected with calcium-binding protein synthesis [57]. Calcium has been shown to primarily affect cell growth, while for phosphorus there is evidence of an increase of osteogenic protein expression in relation to phosphate [58]. When both Ca^2+^ and PO_3_^4−^ were combined in vitro, they activated signaling pathways involving genes typically activated by dexamethasone and its glucocorticoid receptor [55].

Even though HAP has a low solubility rate [50], and thus does not release high amounts of stimulation ions, the HAP construct was shown to have good biocompatibility and osteoinductive properties both in vitro and in vivo [59,60]. These results are in good accordance with our obtained data. The strongest mRNA expression of osteogenic markers RunX2 and collagen I, was only reached in the groups with a hydroxyapatite/collagen I carrier (Figure 4 and Figure 5). Bone extracellular protein osteocalcin was also present in high amounts in these samples (Figure 6). A high solubility rate is not always an advantageous feature. It can cause elevated ion levels and, therefore, pH changes that are unsuitable for cell wellbeing. Based on the results from qPCR, pure Coll scaffold also had a positive effect on osteogenic genes mRNA expression (Figure 4 and Figure 5). However, as is apparent from the immunohistochemical staining of extracellular protein osteocalcin, the strong expression of initial osteogenic transcription factor RunX2 did not lead to protein synthesis (Figure 4 and Figure 6). Based on these results, we can conclude that the material does not provide stimulation strong enough for osteocalcin mRNA translation. Thus, we assume pure Coll scaffold not to be suitable for osteogenic process initiation. Even though we did not observe either significant pH changes or cytotoxic effects, the number of ions released from the materials was probably too high to induce osteogenesis in the case of tricalcium phosphates.

The particle size, as well as the porosity and pore size of the scaffolds, can influence protein adhesion to the surface of certain bioceramic. Cell adhesion can also be influenced [61]. It is reported that particle sizes lower than 100 nm improve protein adhesion [62]. The particle size is more related to the initial adhesion of seeded cells and their proliferation rate than differentiation induction. Particle size is also often connected with the immune system response induction [63]. In general, smaller particles lead to a higher possibility of immune reaction [64]. On the other hand, it has been found that nanoscale hydroxyapatite implanted in vivo results in increased bone formation and stronger interaction between bone cells and the scaffold surface [65]. In this study, we did not observe any common effects of particle size on the osteogenesis of human MSCs. Considering the possible effects on the immune system, it would be adequate to test the macrophage polarization after incubation with the tested materials.

## 4. Materials and Methods

### 4.1. Scaffold Preparation

Commercially available collagen type I was freeze-dried to obtain 100% collagen foam. As additives, 6 types of calcium phosphates (CaP), namely hydroxyapatite (HAP, Reicke company), calcined and noncalcined HAP (Riedel-de-Haen, Germany), alpha tricalcium phosphate (α-TCP, Premier Biomaterials, Nenagh, Ireland), beta-tricalcium phosphate (β-TCP, Fluka, Buchs, Swiss) and calcium polyphosphate (NapolyP transferred with CaCl_2_ to CapolyP, Merck, Prague, Czech Republic), were used. Table 2 describes the composition of prepared scaffolds as well as the type, mean particle size, and the formula of used HAPs. Collagen/CaP scaffolds (weight ratio 1:1) were prepared by the freeze-drying method, according to Sloviková et al. [66], with minor modification. A calculated amount of CaP was slowly added to cold (4 °C) collagen aqueous suspension, with a concentration of 0.5 wt%. Consequently, the mixtures were homogenized and freeze-dried in Martin Christ Epsilon 2-10D lyophilizator (Martin Christ, Osterode am Harz, Germany) at −35 °C under 1 mBar for 15 h, followed by a secondary drying process at 25 °C under 0.01 mBar until decreasing Δp occurred (change in pressure was up to 10%). A post-crosslinking process with a carbodiimides system (the mixture of N-(3-Dimethylaminopropyl)-N′-ethylcarbodiimide hydrochloride (EDC, Sigma-Aldrich) and N-hydroxysuccinimide (NHS, Sigma-Aldrich, Darmstadt, Germany) in ethanol (96%, PENTA, Prague, Czech Republic) with a molar ratio of 2:1) was used. After a 2 h crosslinking process, the scaffolds were twice washed with 0.1 M Na_2_HPO_4_, followed by ultrapure water (three times) for byproduct removal.

### 4.2. Scaffold Morphology

The scaffold morphology was investigated, employing a scanning electron microscope (SEM, Tescan MIRA3, Brno, Czech Republic); Figure 1. All observations were made in the secondary electron emission mode with a high voltage of 10 kV. For better resolution, the scaffolds were coated with a 10 nm gold layer. The pore size of the prepared scaffolds and their distributions were calculated from the SEM images using ImageJ software. The pores were evaluated from 4 different SEM images of each scaffold with a view field of 2.8 mm. The average pore size was calculated using 100 measured values.

### 4.3. Cells and Culture Conditions

Cells were seeded in the third passage in a density of 7 × 10^4^ cells/scaffold. Scaffolds were placed in 48-well culture plates. Human mesenchymal stromal cells from bone marrow (#7500, ScienCell, Carlsbad, CA) were used [67]. 98.44% of cell population had the MSCs phenotype (CD105+, CD45-, CD34-, CD14-, CD19-, CD73+, CD90+, HLA-D-). The cells were cultured in Minimum Essential Medium (51411C, Sigma Aldrich, Darmstadt, Germany) with 10% Fetal Bovine Serum (F75224, Sigma Aldrich, Darmstadt, Germany) and 1% penicillin/streptomycin (15140-122, Invitrogen, Bleijswijk, Netherlands). An osteogenic medium was made by the enrichment of a basal medium with 10 mM β-glycerophosphate (50020, Sigma Aldrich, Darmstadt, Germany), 100 nM dexamethasone (D4902, Sigma Aldrich, Darmstadt, Germany), and 100 µM ascorbate-2-phosphate (A8960, Sigma Aldrich, Darmstadt, Germany). As the negative control group, cells seeded on tissue culture plastics were used. Cells were cultured for 35 days at 37 °C and 5% CO_2_.

### 4.4. Viability Test

The MTS test is used to evaluate the level of cellular metabolism and viability. A yellow MTS substrate (G3581, Promega, Madison, WI, USA)—tetrazolium salt (3-(4,5-dimethylthiazol-2-yl)-5-(3-carboxymehoxyphenyl)-2-(4-sulfophenyl)-2H-tetrazolium)—was incubated with the cells. It was reduced to an insoluble purple formazan by cellular succinate dehydrogenase. To each well, 40 µL of MTS substrate and 200 µL of culture medium was added. The samples were further incubated for two hours in 37 °C and 5% CO_2_. An absorbance of 100 µL of the product was then spectrophotometrically measured at 490 nm, reference wavelength 690 nm. The experiment was carried out in six biological repeats per group.

### 4.5. dsDNA Quantification

After incubation with MTS, substrate scaffolds with cells were rinsed with PBS and transferred to a new Eppendorf tube with 600 µL of lysis buffer. The lysis buffer consisted of 10 mM Tris (T1503, Sigma Aldrich, Darmstadt, Germany), 1 mM EDTA (EDS, Sigma Aldrich, Darmstadt, Germany) and 4 × 10^−4^ % Triton X-100 (T8787, Sigma Aldrich, Darmstadt, Germany). Samples were frozen in the tubes. Samples were further vortexed three times and frozen for two more cycles. Finally, 200 µL of working solution Quant-iT™ PicoGreen dsDNA Assay Reagent (Q33120, Invitrogen, Bleijswijk, Netherlands) was transferred to a black 96-well plate with a transparent bottom. There is a fluorescently labeled probe in the working solution which starts to emit a signal after binding to dsDNA. The fluorescence was measured at an excitation wavelength of 485 nm and emission of 528 nm. The experiment was carried out in six biological repeats per group.

### 4.6. Immunohistochemical Staining of Osteocalcin

The scaffolds were rinsed with PBS three times, and then methanol was added in order to fix the adhered cells. They were kept for 20 min at room temperature; thereafter they were stored at −20 °C. Prior to staining, the scaffolds were washed three times in PBS. The samples were then incubated at room temperature in 0.1% Triton X-100 (T8787, Sigma Aldrich, Darmstadt, Germany) with 1% BSA (SH30574.02, HY Clone, South Logan, Utah) in PBS. After 30 min, the liquid was aspirated and 1% Tween20 (P9416, Sigma Aldrich, Darmstadt, Germany) in PBS was added. The cells were incubated for 20 min at room temperature. Finally, the samples were rinsed with PBS three times. Primary antibody rabbit anti-osteocalcin (T4743, Penninsula Laboratories, San Carlos, CA, USA) was diluted in PBS 1:200 and the samples were incubated overnight. The next day, the samples were rinsed with 0.05% Tween20 (P9416, Sigma Aldrich, Darmstadt, Germany) in three cycles—after 3, 5, and 10 min of incubation in the solution. After one final washing with PBS, a secondary antibody Alexa Fluor 633 (A21070, Life Technologies) in dilution 1:500 in PBS was added, followed by incubation for 50 min at room temperature in the dark. The samples were then rinsed three times in PBS. Hoechst 34580 (H21486, Life Technologies) was diluted at 1:5000 in PBS and added to the samples for 15 min at room temperature. The samples were subsequently rinsed twice in PBS and scanned using a confocal microscope Zeiss LSM 510 DUO (Zeiss, Jena, Germany). Cell nuclei stained with Hoechst 34580 are visualized in blue and osteocalcin in red color. The experiment was carried out in three biological repeats per group. Representative images are presented.

### 4.7. Staining of Nuclei and Cytoplasmic Membrane

The scaffolds were rinsed with PBS three times and then methanol was added in order to fix the adhered cells. They were kept for 20 min at room temperature; thereafter, they were stored at −20 °C. The scaffolds were rinsed three times with PBS. The samples were then incubated at room temperature and in dark with DiOC_6_(3) (D273, Life Technologies, Eugene, Oregon) diluted at 1:700 in PBS. After 45 min, the liquid was aspirated and propidium iodide (P4864, Sigma Aldrich, Darmstadt, Germany) in dilution 1:200 in PBS was added for 10 min. Finally, the samples were washed with PBS three times. For scanning, we used a confocal microscope Zeiss LSM 510 DUO (Zeiss, Jena, Germany). Cell nuclei stained with propidium iodide are visualized in red, cytoplasmic membranes in green color. The experiment was carried out in three biological repeats per group. Representative images are presented.

### 4.8. qPCR

The samples were cut into small pieces and RNA was isolated from them using RNeasy Mini Kit (74104, Qiagen, Hilden, Germany). RNA concentration was measured using NanoQuant Plate (Tecan, Männedorf, Germany) and reader Infinite M200 Pro (Tecan, Männedorf, Germany), the ratio of wavelengths 280/260 nm for purity evaluation. For the next stage, cDNA was synthesized using RevertAid First Strand cDNA Synthesis Kit (K1622, ThermoFisher Scientific, Bleijswijk, The Netherlands). cDNA was synthesized in a 30 µL reaction volume containing 52.8 ng of nucleic acid. Subsequently, polymerase chain reaction was performed on Light Cycler 480 (Roche, Basel, Switzerlands). To each reaction, we added TaqMan™ Gene Expression Master Mix (4369016, Life Technologies, Bleijswijk, Netherlands), PCR Grade Water (03315959001, Roche, Basel, Switzerland), and depending on the desired gene, TaqMan Gene Expression Assay containing both primer and probe. List of assays used: eukaryotic translation elongation factor 1 delta EEF1D (assay ID: Hs02339452_g1, 4331182), runt related transcription factor 2 (assay ID: Hs01047973_m1, 4331182) and collagen type I alpha 1 (assay ID: Hs00164004_m1, 4331182), all of ThermoFisher Scientific. The parameters of each reaction were set as follows: one cycle of pre-incubation at 95 °C for 10 min, 45 cycles of amplification each at 95 °C for 10 s, at 60 °C for 30 s, and at 72 °C for 1 s, finally cooling for one cycle at 40 °C for 30 s. The data were evaluated using the 2^−∆Cp^ method, for each gene relative to the housekeeping gene EEF1D. The experiment was carried out in four biological repeats per group. As a negative control group MSCs seeded on TCP were used.

### 4.9. Statistical analysis

The obtained data were analyzed with a one-way analysis of variance (ANOVA) followed by the Student–Newman–Keuls test with level of significance *p* ≤ 0.001 (marked as the name of the group with an asterisk) and *p* ≤ 0.05 (marked as the name of group). Standard deviation curves are shown in graphs.

## 5. Conclusions

Collagen-based scaffolds with enrichment of different calcium phosphates having the same concentration were prepared and compared on the MSC cell line. This comprehensive study revealed that the different sizes and modifications of the phosphates have not any effect on the biocompatibility of scaffolds. All prepared composite scaffolds were proven to be cytocompatible and suitable for human mesenchymal stromal cell growth in in vitro conditions. Moreover, collagen modified with hydroxyapatite promoted bone extracellular protein osteocalcin synthesis, regardless of modification and morphological properties. However, calcined hydroxyapatite also led to an increased expression of RunX2 and collagen type I mRNA. Thus, the composite of collagen and calcined hydroxyapatite seems to be a promising material for in vivo testing as bone graft substitution and potentially for clinical practice.

## Figures and Tables

**Figure 1 ijms-23-04236-f001:**
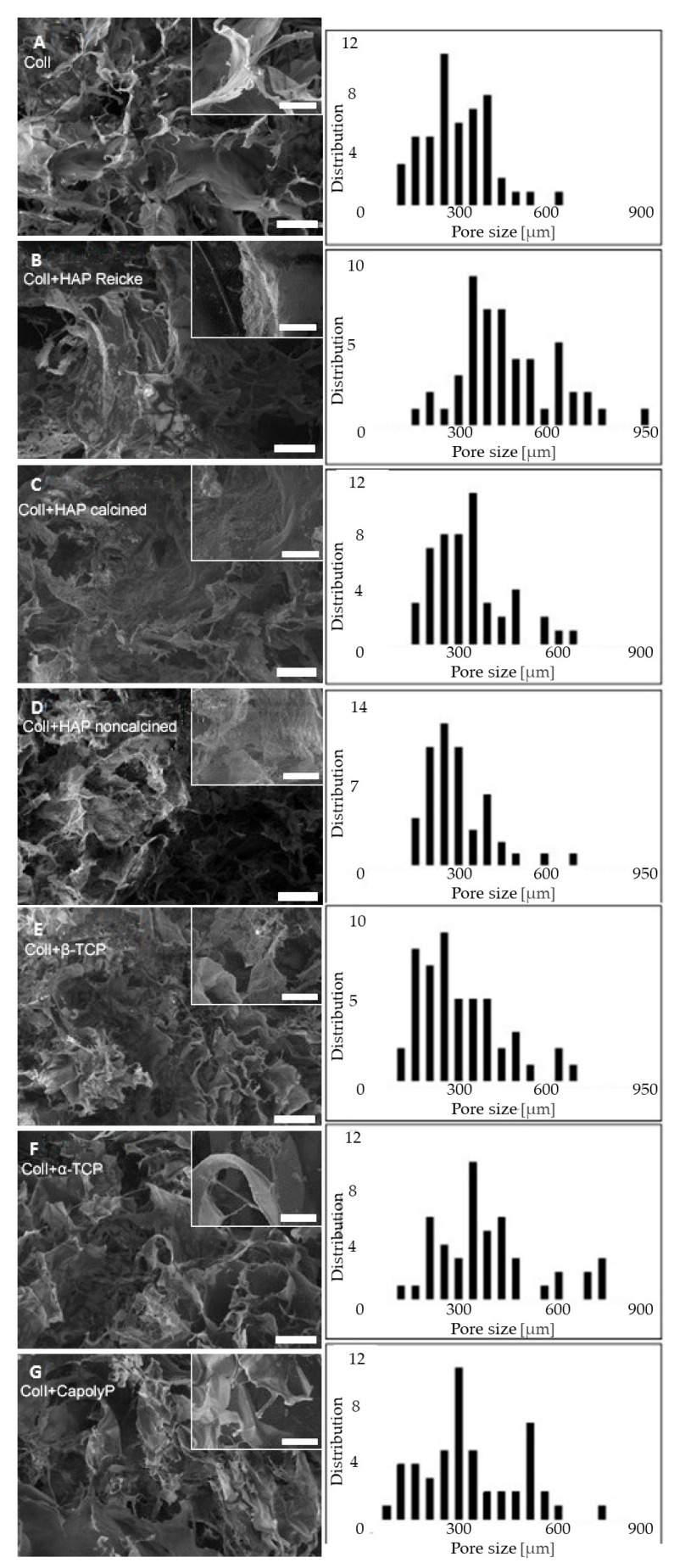
In the left column, there are SEM images of each scaffold, magnification of 222×. Scale bar 200 µm (scaffold morphology), 50 µm, respectively (pore sizes). In the right column, there is the visible pore size distribution of prepared scaffolds. Coll (**A**), ColI+HAP Reicke (**B**), ColI+HAP calcined (**C**), ColI+HAP noncalcined (**D**), ColI+β-TCP (**E**), ColI+α-TCP (**F**) and ColI+CapolyP (**G**).

**Figure 2 ijms-23-04236-f002:**
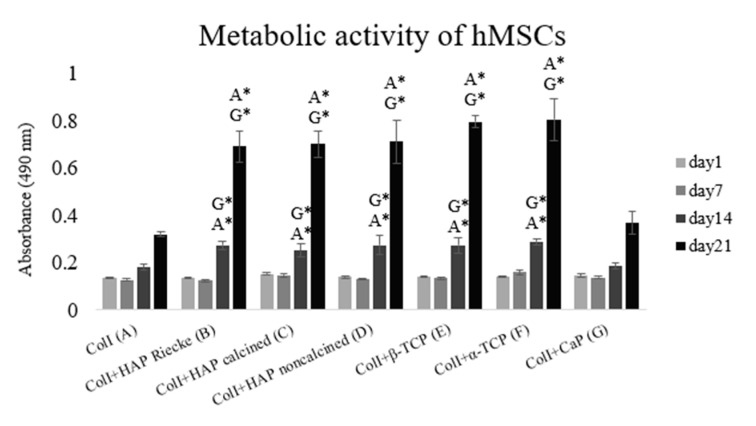
Level of cellular metabolic activity evaluated by MTS test. Cells seeded on: Coll (**A**), ColI+HAP Reicke (**B**), ColI+HAP calcined (**C**), ColI+HAP noncalcined (**D**), ColI+β-TCP (**E**), ColI+α-TCP (**F**) and ColI+CapolyP (**G**). Statistical significance marked with the name of group and asterisk sign for *p* < 0.01.

**Figure 3 ijms-23-04236-f003:**
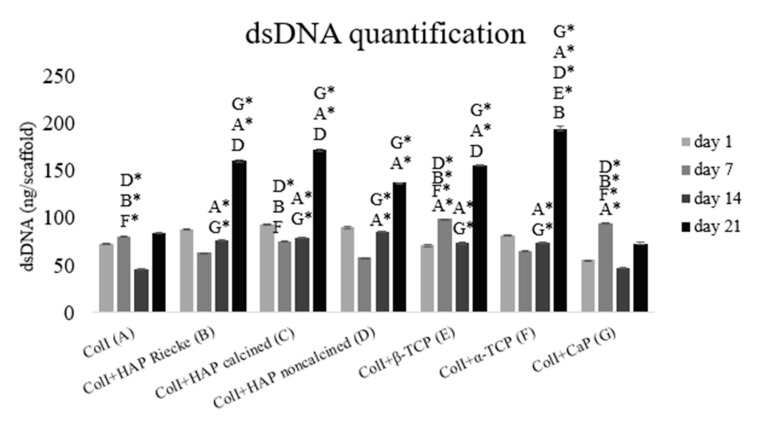
Quantification of cellular dsDNA measured with PicoGreen assay. Cells seeded on: Coll (**A**), ColI+HAP Reicke (**B**), ColI+HAP calcined (**C**), ColI+HAP noncalcined (**D**), ColI+β-TCP (**E**), ColI+α-TCP (**F**) and ColI+CapolyP (**G**). Statistical significance marked with the name of group and asterisk sign for *p* < 0.01.

**Figure 4 ijms-23-04236-f004:**
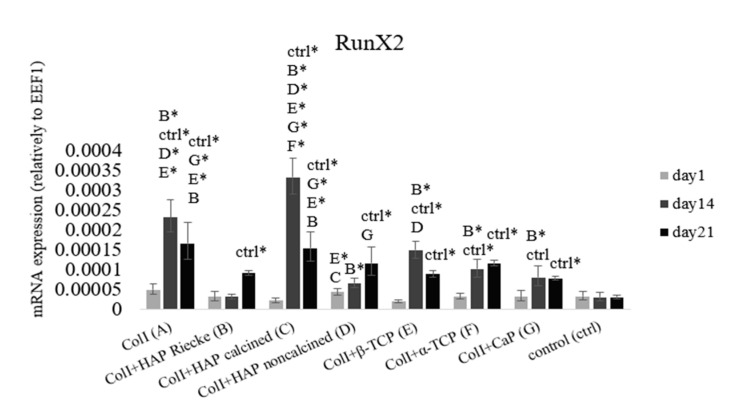
Early osteogenic marker RunX2 mRNA expression relative to EEF1D. Cells seeded on: Coll (**A**), ColI+HAP Reicke (**B**), ColI+HAP calcined (**C**), ColI+HAP noncalcined (**D**), ColI+β-TCP (**E**), ColI+α-TCP (**F**) and ColI+CapolyP (**G**). Negative control group—MSCs on tissue culture plastics. Statistical significance marked with the name of group and asterisk sign for *p* < 0.01.

**Figure 5 ijms-23-04236-f005:**
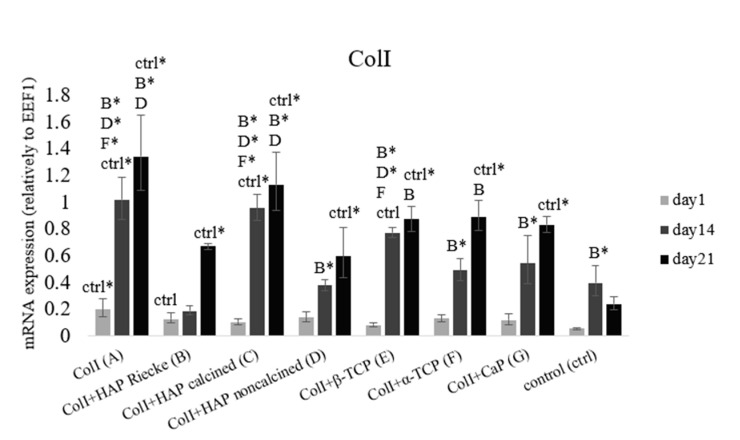
The main extracellular bone protein collagen type I mRNA expression relative to EEF1D. Cells seeded on: Coll (**A**), ColI+HAP Reicke (**B**), ColI+HAP calcined (**C**), ColI+HAP noncalcined (**D**), ColI+β-TCP (**E**), ColI+α-TCP (**F**) and ColI+CapolyP (**G**). Negative control group—MSCs on tissue culture plastics. Statistical significance marked with the name of group and asterisk sign for *p* < 0.01.

**Figure 6 ijms-23-04236-f006:**
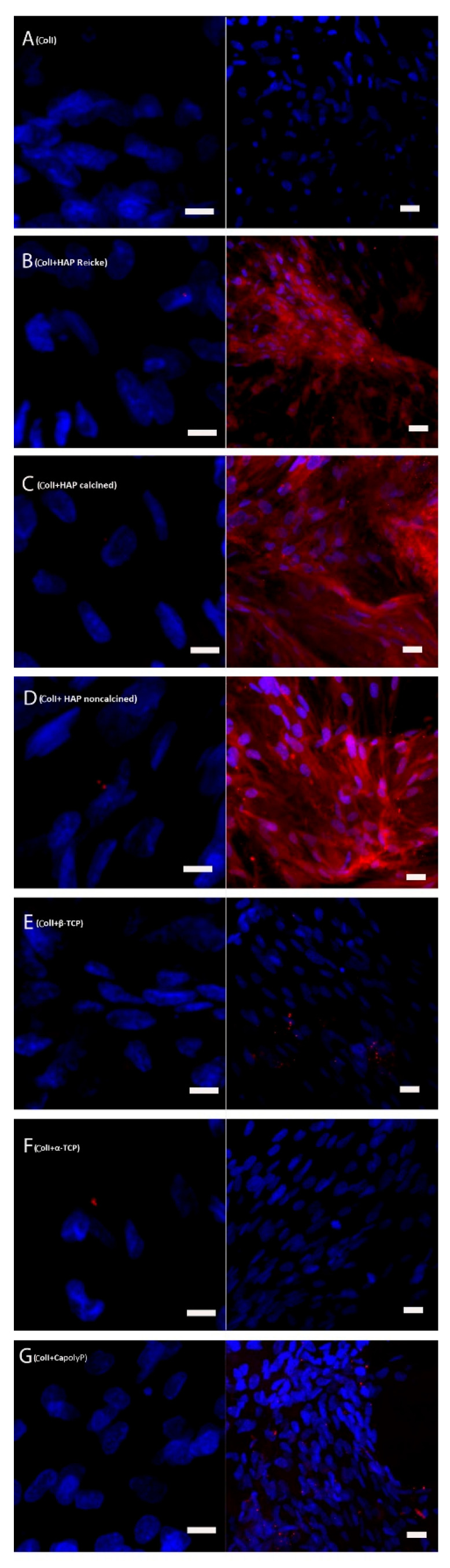
Bone extracellular protein osteocalcin expression on day 21 (left column) and day 35 (right column). Cell nuclei visualized with Hoechst, osteocalcin red. Cells seeded on: Coll (**A**), ColI+HAP Reicke (**B**), ColI+HAP calcined (**C**), ColI+HAP noncalcined (**D**), ColI+β-TCP (**E**), ColI+α-TCP (**F**) and ColI+CapolyP (**G**). Scale bar 10 µm (day 21), 20 µm (day 35).

**Figure 7 ijms-23-04236-f007:**
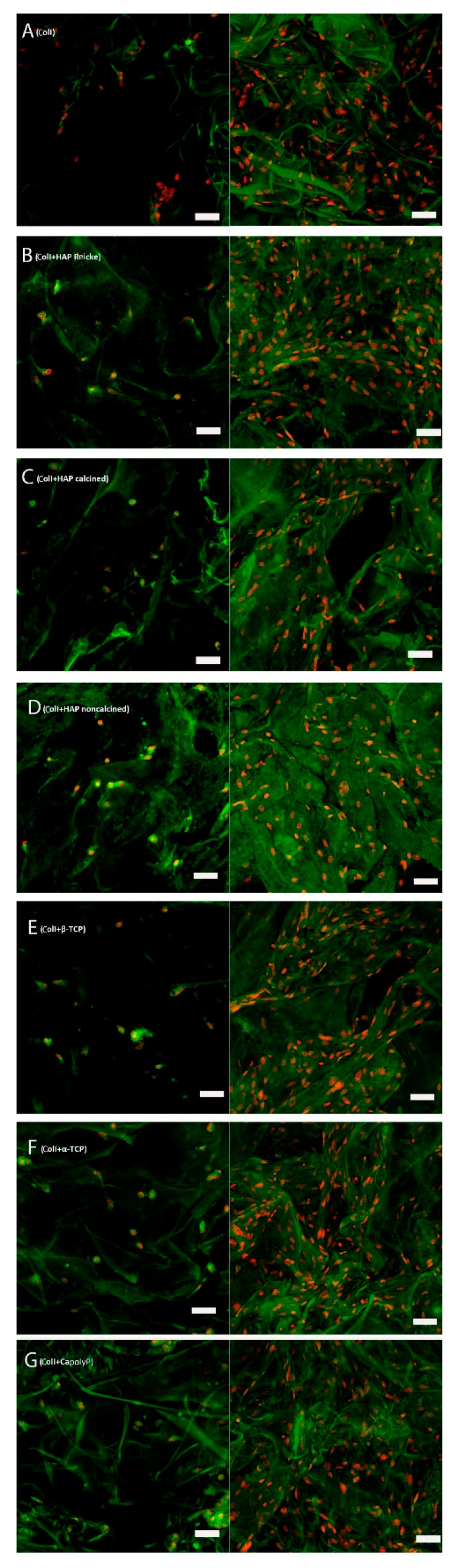
Representative images from confocal microscope on days 1 (left column) and 14 (right column). In red, cell nuclei visualized by propidium iodide; in green, cytoplasm stained by DiOC_6_(3) and scaffold autofluorescence. Scale bar 50 µm. Cells seeded on: Coll (**A**), ColI+HAP Reicke (**B**), ColI+HAP calcined (**C**), ColI+HAP noncalcined (**D**), ColI+β-TCP (**E**), ColI+α-TCP (**F**) and ColI+CapolyP (**G**).

**Table 1 ijms-23-04236-t001:** Pore size of prepared scaffolds measured by ImageJ software.

Sample	Pore Size [µm]
ColI (A)	320 ± 120
ColI+HAP Reicke (B)	490 ± 170
ColI+HAP calcined (C)	360 ± 130
ColI+HAP noncalcined (D)	320 ± 110
ColI+β-TCP (E)	330 ± 150
ColI+α-TCP (F)	460 ± 230
ColI+CapolyP (G)	370 ± 170

**Table 2 ijms-23-04236-t002:** The composition of prepared scaffolds; type; mean particle size and formula.

Sample	Type of Material	Mean Particle Size [µm]	Formula
ColI (A)	Bovine Collagen Type I pure	-	
ColI+HAP Reicke (B)	Hydroxyapatite commercial	30	Ca_10_(PO_4_)_6_(OH)_2_
ColI+HAP calcined (C)	Hydroxyapatite calcined at 1000 °C	0.130	Ca_10_(PO_4_)_6_(OH)_2_
ColI+HAP noncalcined (D)	Hydroxyapatite noncalcined	0.029	Ca_10_(PO_4_)_6_(OH)_2_
ColI+β-TCP (E)	β-tricalcium phosphate	4.21	89 wt% Ca_3_(PO_4_)_2_ 11 wt% Ca_2_P_2_O_7_
ColI+α-TCP (F)	α-tricalcium phosphate	11.01	92 wt% Ca_3_(PO_4_)_2_ 8 wt% Ca_10_(PO_4_)_6_(OH)_2_
ColI+CapolyP (G)	Sodium polyphosphate transferred by CaCl_2_	8.26	Ca(PO_3_)_n_ *n* = 25

## Data Availability

Not applicable.

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
