# Peer review of "Calcined Hydroxyapatite with Collagen I Foam Promotes Human MSC Osteogenic Differentiation"

_ijms, 2022, doi:10.3390/ijms23084236_

Round 1
Reviewer 1 Report
In this article, the authors present collagen foams modified with hydroxyapatite, calcined and uncalcined, or with different calcium phosphates. The behavior of these samples with human
mesenchymal cells was studied.
In order to have a better knowledge of this type of materials, I
have reviewed different articles, among them, the cite 8, with the title: Biologically Inspired Collagen/Apatite Composite Biomaterials for Potential Use in Bone Tissue Regeneration-A Review. In different sections of
this article, the authors describe the results obtained by different researchers, without wishing to be exhaustive, I would like to highlight the following results:
"Mazzoni et al. assessed the biocompatibility, osteoconductive and osteoinductive properties of HA (Pro Osteon 200) and collagen (Avitene) composites using a cell model—mesenchymal stem cells
(hMSC) [70]. Expression of osteogenic genes was analyzed in cells located on the composite. The results showed that such biomaterial has the ability to induce osteogenic differentiation of hMSC, because it induces osteogenic genes and increases matrix mineralization without toxic effects. Other studies also conducted on the cell line have confirmed that the HA/Col composite has osteoinductive properties and is a good tool to accelerate the migration, proliferation and di erentiation of bone tissue cells [71].
The above material was also used in maxillofacial surgery as a kind of scaffolding for the zygomatic bone. The high biocompatibility and the osteoconductive properties of the composite have been confirmed and the low number of postoperative infections has been noted [72]."
- With this paragraph I want to indicate that this type of material is widely known. The same is true for collagen and calcined HA or TCP. When writing the article, the authors have not emphasized the novelty of this work, which is why it has been difficult for me to find it. For example, the conclusions are generic, they do not describe, in my opinion, the novelties provided by this work. For this reason, I believe that the authors should make a greater effort to describe the novelties of this work more adequately.
- On the other hand, in line 13, the authors use the term bioactivity to describe the interaction of the samples with the cells. In my opinion, however, this term is used to describe the ability to precipitate HA when the
samples are immersed in simulated body fluid.
- Finally, I have had difficulties when viewing the figures, although I have used the pdf to enlarge the figures, this point should also be improved.
Author Response
Response to decision letter and list of changes
Manuscript ID: ijms-1653162
Title: Calcined Hydroxyapatite with Collagen I Foam Promotes Human MSCs Osteogenic Differentiation
First, the authors are grateful to the referees for their valuable comments and suggestions that improved the quality of the paper. The answers to specific comments are given below. All the changes are highlighted in yellow.
In this article, the authors present collagen foams modified with hydroxyapatite, calcined and uncalcined, or with different calcium phosphates. The behavior of these samples with human
mesenchymal cells was studied.
In order to have a better knowledge of this type of materials, I have reviewed different articles, among them, the cite 8, with the title: Biologically Inspired Collagen/Apatite Composite Biomaterials for Potential Use in Bone Tissue Regeneration-A Review. In different sections of this article, the authors describe the results obtained by different researchers, without wishing to be exhaustive, I would like to highlight the following results:
"Mazzoni et al. assessed the biocompatibility, osteoconductive and osteoinductive properties of HA (Pro Osteon 200) and collagen (Avitene) composites using a cell model—mesenchymal stem cells (hMSC) [70]. Expression of osteogenic genes was analyzed in cells located on the composite. The results showed that such biomaterial has the ability to induce osteogenic differentiation of hMSC, because it induces osteogenic genes and increases matrix mineralization without toxic effects. Other studies also conducted on the cell line have confirmed that the HA/Col composite has osteoinductive properties and is a good tool to accelerate the migration, proliferation and di erentiation of bone tissue cells [71]. The above material was also used in maxillofacial surgery as a kind of scaffolding for the zygomatic bone. The high biocompatibility and the osteoconductive properties of the composite have been confirmed and the low number of postoperative infections has been noted [72]."
- With this paragraph I want to indicate that this type of material is widely known. The same is true for collagen and calcined HA or TCP. When writing the article, the authors have not emphasized the novelty of this work, which is why it has been difficult for me to find it. For example, the conclusions are generic, they do not describe, in my opinion, the novelties provided by this work. For this reason, I believe that the authors should make a greater effort to describe the novelties of this work more adequately.
Answer:
According to the comments and suggestions, we modified all the text in order to make it clearer and easier to read. We also revised the introduction and added a new description of novelties (lines 59-64: ”In the literature, there are many outstanding studies dealing with collagen and phosphates scaffolds. On the other hand, there is not any comprehensive study comparing different types of phosphates in the same scaffold matrix and concentration as well as by the using same MSC lines. The articles deal only with individual calcium phosphates, but it is not clear which of them is suitable e.g., for in vivo testing and further for clinical practice.” and references to the draft. New information about novelty was also added to the conclusion (lines:526-529 and 534-536). All the changes are highlighted in yellow.
- On the other hand, in line 13, the authors use the term bioactivity to describe the interaction of the samples with the cells. In my opinion, however, this term is used to describe the ability to precipitate HA when the samples are immersed in simulated body fluid.
Answer:
In this case, the use of the word „bioactivity“ was confusing. Thank you for pointing it out. We replaced it with more appropriate terms, e.g. biological activity, biological potential (highlighted in yellow).
- Finally, I have had difficulties when viewing the figures, although I have used the pdf to enlarge the figures, this point should also be improved.
Answer:
The resolution and size of the figures were in accordance with the journal guidelines. However, we adjusted the parameters of all of the figures in order to make them easier to read. We enlarged the labels of experimental groups and axis in graphs.
We also revised English grammar. All the changes are highlighted in yellow.
Please see the attachement.

Reviewer 2 Report
The present paper is a nice report of the effect of scaffold for bone regeneration on human mesenchymal stromal cells. The authors should be commended for the use of a good model and the appropriate scientific tools to investigate the phenomena of interest. I would urge the authors to apply extra care in their presentation of the data. It is not easy for the readers to et their heads around all the different experimental groups. I would revise the introduction, putting more emphasis on the issue that you are investigating (e.g. spending more words on the porosity issue and topographic control of cell function) and introducing right away the different materials that will be tested in the paper. And probably I would revise the figure legends to help readers to always be well aware of what all the different materials are. There are many experimental groups and it is easy to mix them up.
Author Response
Response to decision letter and list of changes
Manuscript ID: ijms-1653162
Title: Calcined Hydroxyapatite with Collagen I Foam Promotes Human MSCs Osteogenic Differentiation
First, the authors are grateful to the referees for their valuable comments and suggestions that improved the quality of the paper. The answers to specific comments are given below. All the changes are highlighted in yellow.
The present paper is a nice report of the effect of scaffold for bone regeneration on human mesenchymal stromal cells. The authors should be commended for the use of a good model and the appropriate scientific tools to investigate the phenomena of interest. I would urge the authors to apply extra care in their presentation of the data. It is not easy for the readers to get their heads around all the different experimental groups. I would revise the introduction, putting more emphasis on the issue that you are investigating (e.g. spending more words on the porosity issue and topographic control of cell function) and introducing right away the different materials that will be tested in the paper. And probably I would revise the figure legends to help readers to always be well aware of what all the different materials are. There are many experimental groups and it is easy to mix them up.
Answer:
Thank you for the time you devoted to reading our manuscript and also for your feedback. According to the comments and suggestions, we modified all the text, all figures, and figure legends in order to make them clearer and easier to read. We enlarged the labels of experimental groups and axis in graphs. We revised the introduction and added information regarding surface characteristics and their influence on cells (lines 50 – 87). We also revised English grammar. All the changes are highlighted in yellow.
Please see the attachement.

Round 2
Reviewer 1 Report
In my opinion, collagen/calcium phosphate materials are widely known, however, the present work, with the modifications made, provides some knowledge, for this reason, I consider that it can be published in the present form